# The Emerging Role of Histotripsy in Liver Cancer Treatment: A Scoping Review

**DOI:** 10.3390/cancers17060915

**Published:** 2025-03-07

**Authors:** Heineken Queen, Sarah F. Ferris, Clifford S. Cho, Anutosh Ganguly

**Affiliations:** 1Research Service, Veterans Affairs Ann Arbor Healthcare System, 2215 Fuller Road, Ann Arbor, MI 48105, USA; heineken@umich.edu (H.Q.); sarahfer@umich.edu (S.F.F.); 2Graduate Program in Immunology, University of Michigan Medical School, Ann Arbor, MA 48109, USA; 3College of Osteopathic Medicine, Michigan State University, East Lansing, MI 48824, USA; 4Department of Cell Biology, Van Andel Institute, Grand Rapids, MI 49503, USA; 5Department of Surgery, University of Michigan, Ann Arbor, MI 48109, USA

**Keywords:** histotripsy, hepatocellular carcinoma, abscopal effect, immunotherapy

## Abstract

Hepatocellular carcinoma (HCC) is an aggressive disease that can be resistant to conventional chemotherapy and radiotherapy. This scoping review explores how histotripsy can potentially be combined with immunotherapy to overcome refractory HCC. By delineating the limitations of current treatment strategies, we provide a comprehensive perspective on the advantages of histotripsy in the management of HCC and offer suggestions on how to maximize its clinical effectiveness.

## 1. Introduction

Despite significant advancements in cancer therapeutics, liver cancer remains one of the leading causes of cancer-related deaths globally [1,2]. The most common liver cancer is hepatocellular carcinoma (HCC). Current treatment modalities, such as transarterial therapy, surgical resection, radiation, chemotherapy, immunotherapy, and thermal ablation, offer limited efficacy, particularly for patients with advanced disease or underlying liver dysfunction [3]. Moreover, many of the currently available therapies are associated with significant risks and side effects. Hence, there remains an ongoing need for novel, non-invasive treatment methods that can effectively target HCC.

Histotripsy is a recently FDA-approved non-thermal, non-invasive, focused ultrasound treatment that may be a solution to this challenge. Early preclinical studies and clinical trials have demonstrated the capability of histotripsy to ablate solid tumors and induce remission with minimal side effects [4,5]. A major advantage of this treatment modality is its suitability for patients with inoperable tumors or those with underlying liver cirrhosis who are not candidates for traditional treatments like surgery or radiation. Additionally, unlike thermal and cryoablation methods, which utilize heat and extreme cold, histotripsy’s non-thermal mechanical tissue disruption has been postulated to preserve the native conformation of tumor antigens, inducing a stronger tumor-directed immune response [6]. Furthermore, preclinical studies have demonstrated the significant immune-stimulating effects of histotripsy [6,7,8,9,10]. This is especially promising in the treatment of HCC, a highly resistant and immunologically “cold” tumor, as it offers the potential to enhance therapeutic outcomes through combination therapies and effectively address tumor metastasis. This review article will discuss the pathophysiology of HCC, assess the challenges of traditional therapeutic approaches, and identify the role of histotripsy, particularly for patients with underlying liver cirrhosis and metastatic tumors. Furthermore, this review will consider the limitations of histotripsy, highlight current areas of ongoing preclinical research, and discuss the directions of future research to further advance histotripsy regimens to improve patient outcomes.

## 2. Methods

The PRISMA checklist for scoping reviews was followed to structure this review [11]. Literature searches were conducted in the PubMed/MEDLINE (National Library of Medicine), Embase (Elsevier), and Scopus (Elsevier) databases, using keywords such as “hepatocellular carcinoma”, “histotripsy”, and “immunotherapy” to identify peer-reviewed articles published in English from 2002 to 2024. In addition, reference lists from selected articles were manually reviewed to identify any relevant studies not included in the initial search. The inclusion criteria covered peer-reviewed articles focusing on early preclinical and clinical trial data related to histotripsy applications for HCC and other cancer types, ensuring translatability. Full texts were retrieved for the articles considered potentially relevant. No further filters were applied. The articles were chosen based on their publication date, prioritizing recent studies, availability in English, and relevance to the following research questions:▪What are the limitations of the currently available treatments for HCC?▪Does histotripsy offer potential benefits compared to the available treatment methods?▪How can histotripsy be further developed to better address the biological complexities that make the treatment of HCC particularly difficult?

All articles were reviewed by both primary authors to ensure their relevance, with any disagreements regarding the selection discussed and resolved by consensus. No data were extracted from the sources for separate charting. To streamline the synthesis, the selected articles were grouped according to the research questions they addressed. No additional data processing was carried out. The findings were organized into categories corresponding to the research objectives, providing a narrative summary of the evidence. A peer-to-peer consultation exercise was conducted to enrich the review by gathering diverse perspectives. The insights from this process helped refine the interpretation of the results and identify potential areas for integrating combinatorial approaches using histotripsy into clinical practice, aiming to address the limitations of the current standard care for HCC.

## 3. Hepatocellular Tumors: Overview

Hepatocellular carcinoma (HCC) is the most common cause of primary liver cancer, accounting for approximately 75–85% of cases worldwide [12]. Arising from hepatocytes, this cancer is highly vascularized and aggressive [13,14,15]. The pathogenesis is divided into either viral or non-viral causes. The two forms of viral hepatitis that have been linked with HCC development are Hepatitis B virus (HBV) and Hepatitis C virus (HCV), while non-alcoholic steatohepatitis (NASH) and alcohol-related liver disease (ALD) are non-viral causes of HCC [16]. In the initial stages, it is often asymptomatic, which, when combined with its potential for metastasis, means that many patients are not diagnosed until later in the disease course. An additional complication is that many patients that develop HCC have underlying liver disease and cirrhosis, which makes the treatment of HCC particularly complicated.

HCC cells are also known to have extensive proliferative capacity, leading to metastasis. Recently, it has been found that the upregulation of cell division cycle-associated 8 (CDCA8), a key protein for chromosomal segregation during mitosis, is associated with HCC metastasis by promoting DNA synthesis and thus tumor cell viability [17]. Additionally, abnormal activation of the growth signaling pathways PI3K/Akt/mTOR and Ras/Raf/MAPK contribute to HCC malignancy [16].

The current treatment options for patients with HCC include surgical resection, transarterial therapy, thermal ablation, radiation, transplantation, chemotherapy, and immunotherapy. These have shown limited success, particularly in treating patients with advanced-stage disease. Surgical resection and transplantation, while potentially curative, are highly invasive. Whereas surgical resection is only available to patients with early disease and good underlying liver function, transplantation is limited by donor organ availability. Transarterial therapies like bland embolization, chemoembolization, and selective internal radiation therapy, as well as ablative therapies like ethanol injection and microwave ablation, are associated with numerous side effects and risks of liver injury [18]. To date, chemotherapy and immunotherapy have been associated with limited response rates. The recent introduction of histotripsy into clinics has offered a potentially effective and highly safe alternative for the treatment of HCC.

### 3.1. Classification of HCC

Classifying HCC based on severity and disease progression is essential for selecting the appropriate treatment method. There are numerous proposed methods for classifying HCC, but the one commonly used in the United States is the American Joint Committee on Cancer (AJCC) classification system. Based on this system, there are four primary stages, which can be further subdivided based on presentation. Stage I consists of primary tumors of <2 cm or >2 cm without vascular invasion, the former being termed stage IA and the latter being stage IB. Stage II is defined as solitary tumors of >2 cm with vascular invasion or multiple tumors, none of >5 cm. Stage III consists of multiple tumors, at least one of which is of >5 cm, or a single tumor of any size that involves a major branch of the portal or hepatic vein or with direct invasion into adjacent organs. Neither stage I, stage II, or stage III has regional lymph node or metastatic involvement, while stage IV has primary tumors of any presentation with regional lymph node and/or metastatic involvement. Another recent guideline for the staging and management of HCC is the Barcelona Clinic Live Cancer (BCLC) prognosis and treatment strategy [19]. This system classifies tumors similarly to the AJCC system; however, an emphasis is placed on how classification guides clinical treatment decisions. Both guidelines highlight that the large majority of current therapies are only effective in the early stages of HCC, when the tumor size is minimal and there is no distant multifocal or metastatic involvement.

Additionally, assessing the patient’s liver function and overall health is critical when choosing the best treatment approach. Many patients with HCC have underlying liver cirrhosis and poor liver function. Thus, treatments that could cause further damage and fibrosis are not viable, as they can lead to liver failure. Furthermore, patients who are not in good health or have significant underlying conditions are less likely to recover from invasive treatments such as surgery and are less capable of handling the side effects from treatments such as radiation.

Lastly, understanding the pathophysiology of HCC, particularly the unique tumor microenvironment and resistance mechanisms, can help guide treatment selection and the development of improved therapeutics. One of the major processes by which HCC develops multi-drug resistance is the overexpression of ATP-binding cassette (ABC) transporters. These ATP-dependent transmembrane proteins pump chemotherapeutics out of the tumor cell, reducing their intracellular concentration [15,20,21,22]. Conversely, the reduced expression of solute carrier proteins (SLCs) can lower the amount of drug effectively taken into the cells [15,20,21]. This makes treatment with chemotherapeutic and immunotherapeutic medications difficult, as attaining sufficient intracellular concentrations to achieve cancer cell death can be challenging. Additional mechanisms such as altered DNA repair, reduced apoptosis, and epigenetic alterations have all been implicated in driving tumor growth and metastasis while reducing HCC’s response to therapy [15,20,21]. Further discussion concerning the tumor microenvironment and its effects on drug delivery and the immune response will be covered when discussing combination therapy.

### 3.2. Current Treatments for HCC and Their Limitations

The standard of care for patients with HCC is dependent on the stage of the disease and the patients’ underlying liver function and overall health. The first-line treatment for early disease is surgical resection. This treatment is potentially curative; however, it is highly invasive and associated with significant risks, and as such, it is only an option for patients who have good underlying liver function and overall health. A review of the American College of Surgeons National Surgical Quality Improvement Project (ACS-NSQIP) showed an increased risk of peri-operative death in patients with fatty (2.8%) or cirrhotic livers (2.6%) who underwent partial hepatectomy for the treatment of HCC compared with those with normal liver texture (0.8%) [23]. This major limitation is because HCC is often not diagnosed until late in the disease progression and often presents in patients with underlying liver disease. Evidence shows that HCC is highly correlated with fibrosis, with over 90% of HCC cases arising in patients with cirrhotic livers [24].

Alternative liver-directed treatments for HCC include stereotactic body radiation therapy, transarterial therapies, and ablation. Stereotactic body radiation therapy can be used in patients with later-stage disease; however, there is a significant risk of radiation-induced liver disease, leading to increased fibrosis and the need for transplant. Hence, it is often not a good option for patients who already present with cirrhosis. Thermal ablation is also an alternative; however, it is only viable for small tumors that are not in close proximity to major blood vessels due to the risk of thermal damage, limiting its therapeutic efficacy. Transarterial therapies work by cutting off the blood supply to the tumor by delivering chemotherapeutics or radioactive microspheres directly to the feeding arteries. This treatment is an option for patients with larger tumors; however, it is not effective in treating metastatic disease. Lastly, liver transplantation is an option to treat HCC. However, liver transplantation is not always readily available and is restricted to patients with early-stage disease due to the risk of recurrence following transplantation in patients with late-stage disease.

Several chemotherapeutic drugs have also received FDA approval for the treatment of localized and advanced HCC. Sorafenib is currently the standard chemotherapeutic agent for advanced HCC. It is classified as a multikinase inhibitor that inhibits tumor cell proliferation by targeting the Raf/MEK/ERK signaling pathway and angiogenesis by blocking vascular endothelial growth factor receptors (VEGFRs) and platelet-derived growth factor receptors (PDGRF) [21]. Unfortunately, HCC can develop resistance to Sorafenib, warranting alternative approaches.

Immune checkpoint inhibitors that target the inhibitory proteins like programmed death-ligand 1 (PD-L1) on cancer cells and cytotoxic T-lymphocyte antigen 4 (CTLA-4) on T cells have received FDA approval for the treatment of advanced HCC, including Atezolizumab, Tremelimumab, Pembrolizumab, and Nivolumab [25]. Due to acquired resistance of advanced HCC, a combinatorial approach using Nivolumab and Ipilimumab for patients who were previously treated with Sorafenib was granted accelerated FDA approval. However, this combination strategy is associated with increased risk and rates of toxicity [26].

Patients with advanced disease or disease that is not amenable to liver-directed treatment options may be eligible for the combination systemic therapy regimen of Atezolizumab and Bevacizumab. Atezolizumab is a monoclonal antibody that binds to PD-L1 on cancer cells to enhance T-cell function, and Bevacizumab is a monoclonal antibody that binds circulating VEGF to inhibit angiogenesis. The Phase 3 IMbrave150 clinical trial (NCT03434379) showed improved overall survival (OS) and progression-free survival (PFS) with this combination regimen compared to Sorafenib, which has been the historic standard of care [27]. While this combination proved more effective, the 12-month survival was still only 67.2%, a modest increase of 12.6% over the Sorafenib [27]. Table 1 summarizes the advantages and limitations of different treatment modalities for HCC.

## 4. Histotripsy

Recently FDA-approved to treat liver cancers, histotripsy is a novel technology that utilizes non-thermal, high-intensity focused ultrasound pulses to non-invasively ablate targeted regions of tissues into acellular debris [28]. Histotripsy works through a process known as cavitation—the formation and then collapse of bubbles within a medium. The acoustic waves emitted during histotripsy generate negative pressures within the target tissue, which drive the formation, growth, and collapse of these bubbles. The process by which this occurs and the subsequent cavitation pattern generated vary based on device parameters including pulse length, peak power, and pulse repetition frequency (PRF) [28,29]. Currently, there exist three major classifications of histotripsy, which vary based on the aforementioned parameters; these include shock-scattering histotripsy, boiling histotripsy, and intrinsic threshold histotripsy. Many of the current methods used in experimental studies are variations or combinations of these three basic approaches. Although the mechanisms behind achieving cavitation vary, all the approaches ultimately produce a similar outcome: the disruption and liquefaction of the target tissue.

### 4.1. Types of Histotripsy

Shock-scattering was the first histotripsy method developed. It utilizes short pulses (5–25 µs) in sequence to generate peak negative pressures between 15 and 25 MPa [30]. This drives the nucleation and cavitation of bubbles within the target tissue, which subsequently generates shockwaves, further destroying the surrounding tissue. This process generates a cavitation cloud that is typically spherical but can be hemispheric or concave in form [31]. A major limitation of this method is the inability to precisely control the shockwaves and the resulting cloud-shape growth, which can lead to less precise ablation margins compared with other forms of histotripsy [31].

Boiling histotripsy uses longer (1–30 ms) pulses to generate peak negative pressures between 10 and 18 MPa [30]. The longer pulses cause the target region to heat to the point of boiling. The bubble generated from this boiling is then capable of generating shockwaves similar to shock-scattering histotripsy. The shape of the subsequent cavitation cloud resembles that of a tadpole, with the head being the site of the boiling and the tail being the site of the subsequent shockwaves that are generated [31]. This process occurs so rapidly as not to cause thermal damage to the surrounding tissue; hence, boiling histotripsy can also be considered a non-thermal focused ultrasound technique. Like shock-scattering histotripsy, a disadvantage to this method may be the inability to precisely control the ablation margins. However, a potential benefit is the ability of boiling histotripsy to destroy fibrosis, a process that will be discussed further in subsequent sections.

Intrinsic threshold histotripsy utilizes a single short (1–2 µs) pulse to generate a peak negative pressure of >27 MPa [30]. This method only uses a single pulse and thus does not generate shockwaves. This means that the cavitation cloud is often smaller, and repeated cycles are needed to completely ablate the targeted region [31]. However, this method is also the most precise and allows for the cleanest margins compared with the other methods described.

The *HistoSonics* Edison^®^ (Ann Arbor, MI, USA) platform is an FDA-approved histotripsy device that utilizes cavitation cloud histotripsy, a form of histotripsy that combines shock-scattering and intrinsic threshold histotripsy. This system has shown significant promise in treating HCC in clinical trials. However, one limitation of this system is that it relies on ultrasound (US) guidance to detect the area for treatment. Not all liver tumors are well-visualized with US, meaning some patients are not candidates for this treatment. Additional limitations include the risk of injury to adjacent structures and sound interference from the lungs. Ongoing technological developments with computed tomography fusion may enhance the ability to target lesions.

### 4.2. Parameter Studies

Studies have shown that the composition of the medium affects the efficacy of histotripsy. To understand this, it is important to understand the terms. The nucleation threshold refers to the energy required for nucleation—bubble generation—to occur, whereas the intrinsic threshold refers to the energy (or peak negative pressure) at which cavitation—bubble collapse—occurs. Both the nucleation threshold and the intrinsic threshold are dependent on the target medium. Vlaisavlejevich et al. showed that media with higher stiffnesses require higher peak negative pressure to achieve cavitation, whereas those with lower stiffnesses will cavitate with lower inciting peak negative pressures [31]. In addition, temperature influences cavitation, with media at higher temperatures requiring less energy (or lower peak negative pressures) compared with those at lower temperatures [31]. These parameters are important to consider when treating different tissues, as each tissue has a unique composition.

As mentioned previously, many patients with HCC have significant liver fibrosis, which can increase the stiffness of the liver and alter the efficacy of histotripsy. A recent study from Joung et al. showed that boiling histotripsy can effectively treat fibrosis in animal models of liver fibrosis [32]. It is thought that this method is more effective than the other forms of histotripsy because the heating of the target tissue lowers the nucleation and intrinsic threshold, improving the efficacy of cavitation. In addition, the heat can also help prevent further scar formation [32,33]. Further investigations need to be performed to ascertain the safety of this protocol and how it can be employed with current treatment regimens to address both HCC and fibrosis concurrently.

Lastly, histotripsy is known to stimulate the immune system and alter the tumor microenvironment (TME). Preclinical data indicate that the degree to which this occurs appears to be variable based on the modality and power used during histotripsy [9]. Ongoing investigation into changes to the TME and the immune response needs to be cognizant of this variable, and further work should be completed to identify what modality/parameters are best for stimulating the optimal response.

### 4.3. Preclinical Data

The non-ionizing and non-thermal properties of histotripsy offer an advantage to alternative procedures such as radiotherapy and thermal ablation, which could result in radiation toxicities and liver damage [34]. Numerous preclinical studies have been conducted to evaluate the safety and efficacy of histotripsy. Sukovich et al. was one of the first groups to examine the feasibility and safety of histotripsy, using the porcine brain as a model. Therein, they determined the capability of histotripsy to generate concise lesions in the brain cortex, which did not lead to adverse effects such as hemorrhage or substantial edema [35]. Qu et al. first showed that histotripsy, in addition to simulating intratumoral immune cell infiltration into melanoma and hepatic tumors, can also promote abscopal effects that inhibit metastasis and enhance the efficacy of immune checkpoint inhibitors [6]. In the same study, the group found that immunomodulation was associated with the release of high-mobility group box 1 protein (HMGB1) and membrane translocation of calreticulin (CRT), both of which are known as damage-associated molecular proteins (DAMPs) that trigger immune responses by inducing immunogenic cell death (ICD) [36]. Additionally, Iwanicki and colleagues demonstrated that histotripsy can induce tumor apoptosis through the activation of caspase-3 via increased TNFα levels, effectively reducing tumor hypoxia in a neuroblastoma model [37].

### 4.4. Clinical Trials

The first human clinical trial of histotripsy for liver cancers, named the THERESA Study (NCT03741088), resulted in the establishment of histotripsy’s efficacy in destroying targeted tissue without harmful device-related effects [38]. Briefly, that study consisted of eight patients who had end-stage liver tumors and were treated with a prototype histotripsy system developed by *HistoSonics*, Inc. (Ann Arbor, MI, USA). Eight patients who were enrolled in that study had unresectable end-stage multifocal liver tumors: five patients had colorectal liver metastases, one had breast cancer metastasis, another had cholangiocarcinoma metastasis, and one had hepatocellular carcinoma. Two out of the eight patients treated revealed decreased tumor markers eight weeks post-treatment, as well as evidence of abscopal effects [38]. In addition to meeting that study’s primary endpoint to safely create a focal ablation zone in the liver, assessed by MRI 1 day post-treatment, its secondary endpoint of identifying device-related adverse effects was also a success—liver function was assessed at different time points (1 week, 1 and 2 months) post-treatment; while elevated transaminase levels were found in four patients, they returned to baseline 1 week after histotripsy. The success of that trial prompted the onset of clinical studies using histotripsy. Following the THERESA Study, the HOPE4LIVER clinical trial was conducted (NCT04572633), which evaluated the safety and efficacy of histotripsy for primary (18 participants) and metastatic (26 participants) liver cancers [5]. While major procedure-related complications were observed (three of forty-four participants), 95% (42 of 44) of the treated tumors achieved technical success within 36 h post-procedure. The technical success and minor procedure-related complications reported in that trial led to histotripsy’s FDA approval to treat liver cancers in October 2023. Currently, the HOPE4KIDNEY (NCT05820087) and GANNON (NCT06282809) clinical trials are ongoing to investigate histotripsy’s potential to treat primary solid renal tumors and pancreatic adenocarcinoma, respectively.

### 4.5. Activation of the Adaptive Immune Response

Histotripsy’s ability to stimulate the immune response provides a means to sensitize immunologically cold tumors to immunotherapy. Preclinical studies have shown that histotripsy can induce intratumoral expression of PD-L1 [7]; thus, one treatment strategy that would ideally be beneficial is combining histotripsy with ICIs such as anti-PD-L1 and anti-CTLA-4 to potentiate their efficacy. Pepple et al. demonstrated that histotripsy alters the immune landscape of the tumor microenvironment, initiating cancer cell necroptosis that primes an adaptive immune response [8]. Activating anti-tumor lymphocytes through necroptosis provides an additional mechanism that may contribute to the priming of anti-tumor CD8+ T cells, which may further augment the efficacy of immunotherapy. Furthermore, they also showed that the abscopal effects observed post-histotripsy were associated with ferroptosis, which is an iron-dependent type of programmed cell death resulting from accumulation of lipid peroxides [8]. Tumor ferroptosis was recently found to be regulated by effector CD8+ T cells via the expression of two solute carrier proteins, SLC3A2 and SLC7A11, wherein their downregulation leads to impaired uptake of cystine by the cancer cells to trigger ferroptosis [39]. Indeed, Pepple and colleagues observed a spatiotemporal dynamic between tumor ferroptosis and CD8+ T cells. Notably, the phenotypic characteristics of the immune milieu of the histotripsy-treated and abscopal tumors revealed that, at later time points, there is an overlap in the immune response pathways that are upregulated. Taking this into account, it is possible that there exists a window in which immunotherapy can be administered post-histotripsy to generate a more robust anti-tumor immune response. Moreover, Hendricks-Wenger et al. showed that histotripsy treatment of in vivo and in vitro models of pancreatic adenocarcinoma induced immune-stimulating factors that activated the innate immune response, resulting in increased tumor-progression-free time and longer survival in mice [10].

## 5. Potential Therapeutic Strategies with Histotripsy

Liver-directed therapies such as surgical resection and transarterial therapy, although they can increase patients’ overall survival, become ineffective for metastatic HCC. On the other hand, systemic therapies are often associated with unwanted side effects and toxicities that do not necessarily target tumor cells. Thus, combinatorial treatments are becoming a more desirable approach, especially for tumors that are resistant to monotherapy. The ability of histotripsy to induce a systemic anti-tumor immune response based on preclinical studies reveals its potential clinical applications for the treatment of liver cancers.

### 5.1. Histotripsy and Nanotechnology

A recent investigation by Edsall et al. into combining histotripsy with nanoparticle technology could be the solution to the visualization requirement of the current Edison^®^ (*HistoSonics*, Inc., Ann Arbor, MI, USA) system. By utilizing nanoparticles that traffic directly to the tumor, they hypothesize that tumors can be treated without the need for direct visualization [40]. Experiments performed in agarose phantoms showed similar efficacy in ablation to current histotripsy protocols using dual-frequency pulsing nanoparticle-mediated histotripsy [40]. Further investigations need to be performed in vivo to assess the efficacy of nanoparticle tracking and potential side effects to ascertain the viability of this solution.

### 5.2. Histotripsy and Immunotherapy

The refractory nature of some cancers to immunotherapy is due in part to the inaccessibility of tumor-associated antigens (TAAs) for antigen-presenting cells (APCs). Histotripsy ablation has been posited to preserve the native conformation of TAAs, allowing for augmented presentation of APCs, such as dendritic cells and macrophages, to effector lymphocytes to elicit a more robust immune response [6]. Indeed, several researchers have shown histotripsy’s ability to stimulate tumor-directed immune responses [6,7,8,9,10]. Generally, one of the mechanisms of cancer resistance to immunotherapy is a lack of immune infiltration due to tumors’ ability to inhibit T-cell activation by reducing their antigen expression. Histotripsy inducing DAMPs may provide an alternate mechanism by which cytotoxic T lymphocytes can circumvent this dampening of tumor-associated and tumor-specific antigen expression by cancer cells, leading to increased immunogenicity. The evidence of an abscopal effect further demonstrates that local histotripsy ablation results in a systemic immune response; thus, metastatic and/or untreated tumors can also be sensitized after histotripsy. These findings suggest that histotripsy can promote the engagement of APCs and CD8+ T lymphocytes, which leads to the proliferation and expansion of tumor-specific cytotoxic lymphocytes (as shown in Figure 1). Hence, priming the immune system with histotripsy to yield a more favorable environment for immunotherapy would be an ideal approach that could result in tumor clearance.

Based on preclinical studies that have demonstrated histotripsy’s ability to promote immunomodulation, tumors could potentially be initially treated with histotripsy, followed by administration of immunotherapy (such as cancer vaccines, monoclonal antibodies, etc.) to further augment the anti-tumor response. However, the optimal window in which immunotherapy should be delivered following histotripsy to achieve the optimal immune response remains to be defined. Additionally, certain tumor types may require distinct histotripsy parameters to stimulate anti-tumor responses; thus, parameter studies are necessary to establish specific guidelines to treat different tumors accordingly. Of note, preclinical studies have yet to identify whether histotripsy parameters can be standardized across distinct types of cancers.

## 6. Discussion

In this scoping review, we selected 38 studies addressing the emerging role of histotripsy in the treatment of hepatocellular carcinoma (HCC). Emphasis was placed on recent (within the last 5 years) publications (75%). One possible limitation of our search is that we screened for English-only publications, possibly missing relevant studies published in other languages. Synthesis of these materials revealed that histotripsy, a novel non-invasive technique recently FDA-approved to treat liver cancers, offers several alternatives over current treatment modalities. This review summarizes the current treatments of HCC and offers insight into histotripsy’s potential to address the limitations of the current standard-of-care regimen and suggests that more research should be directed toward improving current histotripsy regimens, including the potential of combining therapy with immunotherapy.

The varying treatment responses of several cancer types to immunotherapy warrant the need for additional, if not an alternative, approach to resistant cancers. Histotripsy is a relatively new method of tumor ablation therapy that was recently FDA-approved to treat liver cancers. Guided by real-time imaging such as ultrasound or MRI, histotripsy utilizes focused ultrasound to non-invasively ablate targeted regions of tissues into acellular debris [4,28]. To date, several preclinical studies have shown that histotripsy can promote anti-tumor immune stimulation, making it a desirable strategy to potentiate the efficacy of immunotherapy. Preclinical data showing reduced tumor hypoxia after histotripsy via the activation of specific immune cellular death pathways also provide another mechanism by which histotripsy sensitizes tumors to immunotherapy by providing a microenvironment more conducive to robust T-cell effector functions.

## 7. Conclusions

Despite histotripsy’s safety and efficacy in preclinical and clinical studies thus far, certain tissues prove to be a challenge. One limitation that warrants further investigation includes establishing a method to precisely target less accessible tumors due to their location. Parameter studies to determine the optimal pulses to effectively ablate different tumor types are also an area of active research. Despite these limitations, histotripsy’s ability to stimulate the immune response makes combination therapy of histotripsy with immune checkpoint inhibitors a promising treatment strategy for refractory cancers. The evidence of the abscopal effect observed in clinical and preclinical studies further suggests that local tumor ablation induces systemic immunomodulation, additionally contributing to enhanced anti-tumor responses that may synergistically work with immunotherapy. Considering that this combinatorial approach with histotripsy potentially leads to better prognosis for cancer patients, it will be pivotal to translate these findings into clinics to effectively optimize the potency of immunotherapy.

Thus far, histotripsy has shown clear advantages over traditional therapeutic efforts for the treatment of liver cancers; however, additional studies need to be carried out to optimize histotripsy regimens to address liver fibrosis, metastasis, and recurrence. This review article covers the currently FDA-approved histotripsy protocol and highlights current areas of research that could be applied to improve the clinical regimen, including choosing an appropriate method, adjusting doses as necessary, implementing nanoparticle technology, and utilizing combination therapy. The active areas of investigation include dissecting the mechanisms by which this non-invasive modality can reprogram the TME to stimulate the immune response through DAMPs or ICD pathways to confer a conducive environment for immunotherapy to achieve optimal efficacy in refractory and/or late-stage liver cancers.

## Figures and Tables

**Figure 1 cancers-17-00915-f001:**
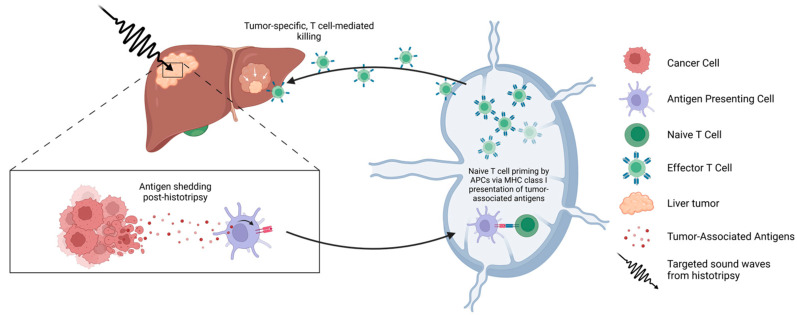
Abscopal effect observed post-histotripsy. Treating liver cancer with histotripsy exposes tumor-associated antigens that are taken up by antigen-presenting cells (APCs). APCs traffic (black arrow) to the spleen, priming and activating T lymphocytes. Effector T cells can recognize and act on distal, untreated tumors, suggesting histotripsy promotes tumor-specific T-cell-mediated killing.

**Table 1 cancers-17-00915-t001:** Comparison of current HCC treatment modalities.

Treatment Modality	Potentially Curative	Minimally Invasive	Minimal Side Effects	Can Treat Any Stage
Surgical resection	✔			
Transarterial therapies		✔		
Combination of Atezolizumab and Bevacizumab		✔		✔
Stereotactic body radiation therapy		✔		✔
Thermal ablation		✔		
Liver transplant	✔			
Chemotherapy		✔		✔
Histotripsy	✔	✔	✔

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
