# Peer review of "The Emerging Role of Histotripsy in Liver Cancer Treatment: A Scoping Review"

_cancers, 2025, doi:10.3390/cancers17060915_

Round 1
Reviewer 1 Report
Comments and Suggestions for Authors
This review summarized current treatments of HCC, and offered insight into histotripsy’s potential to address the limitations of current standard treatment possibly by combining it with immunotherapy. However, there was no enough evidence to show histotripsy have the obvious advantage over the current local treatment of HCC such as thermal ablation, cryoablation, HIFU etc. the conclusion needs to be careful. Also it will be convinced to provide the real cases of the historipsy treating HCC. The review is too long and need to be compressed.
Author Response
Thank you for your insightful comment. We have revised the discussion section according to the reviewer’s suggestion and replaced the term "advantages over other treatment modalities" with "alternatives." Additionally, we have shortened the conclusion and streamlined the discussion section to reduce the overall length. However, in response to Reviewer 3's suggestion, we have restructured the review as a scoping review, incorporating the methodology and selection criteria.
Reviewer 2 Report
Comments and Suggestions for Authors
I appreciate the opportunity to review the paper titled “The emerging role of histotripsy in liver cancer treatment: Current insights and future perspectives."
The study sought to summarizes current treatments of HCC, and offers insight into histotripsy’s potential to address the limitations of current standard of care regimen possibly by combining it with immunotherapy. This review comprehensively delineates potential benefits and downsides of new treatment strategy, histotripsy, for HCC management. Overall, it is well-written and suitable for publication to Cancers. However, I have a few apprehensions and would like to bring them up to potentially strengthen the study.
· Section 2.1: To offer a broader view of treatment flowchart for HCC, I would suggest mentioning recent BCLC classification and management options according to the criteria.
· Related to this, I wonder how histotripsy will change the paradigm of this treatment strategy. I would be grateful if the authors provide their thought on this.
· Section 2.2: I recommend including a citation in Line 138 to provide readers with a clearer understanding of the morbidity and mortality associated with hepatectomy.
· Section 3.4: If possible, could the authors provide several results of clinical trials using numbers, such as study cohort, efficacy, complication rates.
Thank you for considering my comments.
Author Response
Comment: Section 2.1: To offer a broader view of treatment flowchart for HCC, I would suggest mentioning recent BCLC classification and management options according to the criteria. Related to this, I wonder how histotripsy will change the paradigm of this treatment strategy. I would be grateful if the authors provide their thought on this.
Answer: Thank you for the insightful comment, now in section 3.1 we have incorporated the text " Another recent guideline for staging and management of HCC is the Barcelona Clinic Live Cancer (BCLC) prognosis and treatment strategy [19]. This system classifies tumors similarly to the AJCC system, however an emphasis is placed on how classifications guides clinical treatment decisions. Both guidelines highlight that the large majority of current therapies are only effective in the early stages of HCC when the tumor size is minimal and there is no distant multifocal or metastatic involvement. "
Section 2.2: I recommend including a citation in Line 138 to provide readers with a clearer understanding of the morbidity and mortality associated with hepatectomy.
Thank you for the insight we have included in section 3.2 “A review of the American College of Surgeons National Surgical Quality Improvement Project (ACS-NSQIP) showed an increased risk of peri-operative death in patients with fatty (2.8%) or cirrhotic livers (2.6%) who underwent partial hepatectomy for treatment of HCC compared to those with normal liver texture (0.8%) in section 3.2”
Section 3.4: If possible, could the authors provide several results of clinical trials using numbers, such as study cohort, efficacy, complication rates.
We have provided several clinical trial examples in section 3.4 with cohort efficacy and complication rates.
PS: Please note that the section number has been changed from 2.1 to 3.1 and accordingly because of reformatting the manuscript as per reviewer 3's suggestion
Reviewer 3 Report
Comments and Suggestions for Authors
The authors presented a narrative review that summarizes the current treatments of HCC, and offers insight into histotripsy’s potential to address the limitations of current standard of care regimen possibly by combining it with immunotherapy. While the topic is coherently and clearly presented, the manuscript currently is in the textbook format (i.e. reminds of a chapter in a teaching material) and does not meet the modern requirements for the topical/relevant evidence based review providing insights and recommendations for the clinicians. Therefore, the major revision is needed to make it either into systematic or scoping review, so it meets the current PRISMA standards required for publication. See for the details: https://www.prisma-statement.org/scoping.
Author Response
This is an excellent suggestion, We have included the methodology. Additionally, we have formatted the manuscript according to the PRISMA guidelines. Major changes are highlighted
4o
Reviewer 4 Report
Comments and Suggestions for Authors
The paper presents a new method, with the principles first presented 5 years ago, but the first papers presenting results published in 2024.
It is a well balanced review, starting with the classification and current treatments for HCC and their limitations.
The mechanism of histotripsy and the types of histotripsy are detailed, with their limitations and specific features.
The preclinical data and clinical trials on the topic are presented in detail.
The potential therapeutic strategies with histotripsy are presented, consisting in combining histotripsy with nanotechnology and immunotherapy.
The review is clear, comprehensive and of relevance to the field and has the merit of containing all the information available at the moment.
I found no similar review.
The references are appropriate, up-to-date and contain 36 titles. I found no self-citation.
The table gives a good overview on the current HCC treatment modalities.
The conclusions underline the limits of the method but also the main advantaged - histotripsy’s ability to stimulate the immune response and the evidence of the abscopal effect.
In my opinion the paper fits the journal and the language is correct and understandable.
I recommend the paper to be accepted.
Author Response
Thank you for your insightful comments and recommendation for acceptance.
Round 2
Reviewer 1 Report
Comments and Suggestions for Authors
the revised paper is improved and can be acceptable.
Reviewer 3 Report
Comments and Suggestions for Authors
I would like to thank the authors for all the efforts to improve the quality of the manuscript during the review process. The main issues were resolved and the overall structure and organization of the paper have been greatly improved. It could be accepted in current form, although many readers would appreciate if the information was presented even in more concise and compact format.